# Emerging Biomarkers and the Changing Landscape of Small Cell Lung Cancer

**DOI:** 10.3390/cancers14153772

**Published:** 2022-08-03

**Authors:** Anna Keogh, Stephen Finn, Teodora Radonic

**Affiliations:** 1Department of Histopathology, St. James’s Hospital, D08 NHY1 Dublin, Ireland; stephen.finn@tcd.ie; 2Department of Histopathology and Morbid Anatomy, Trinity Translational Medicine Institute, Trinity College Dublin, D08 HD53 Dublin, Ireland; 3Department of Pathology, Amsterdam University Medical Center, VUMC, University Amsterdam, 1081 HV Amsterdam, The Netherlands; t.radonic@amsterdamumc.nl

**Keywords:** small cell lung cancer, ASCL1, NEUROD1, POU2F3, inflamed subtype, DLL3, SLFN11, PARPi, immunotherapy, YAP1

## Abstract

**Simple Summary:**

Small cell lung cancer (SCLC) is an aggressive cancer representing 15% of all lung cancers. Unlike other types of lung cancer, treatments for SCLC have changed very little in the past 20 years and therefore, the survival rate remains low. This is due, in part, to the lack of understanding of the biological basis of this disease and the previous idea that all SCLCs are the same. Multiple recent studies have identified that SCLCs have varying biological activity and can be divided into four different groups. The advantage of this is that each of these four groups responds differently to new treatments, which hopefully will dramatically improve survival. Additionally, the aim of these new treatments is to specifically target these biological differences in SCLC so normal/non cancer cells are unaffected, leading to decreased side effects and a better quality of life. There is still a lot unknown about SCLC, but these new findings offer a glimmer of hope for patients in the future.

**Abstract:**

Small cell lung cancer (SCLC) is a high-grade neuroendocrine malignancy with an aggressive behavior and dismal prognosis. 5-year overall survival remains a disappointing 7%. Genomically, SCLCs are homogeneous compared to non-small cell lung cancers and are characterized almost always by functional inactivation of RB1 and TP53 with no actionable mutations. Additionally, SCLCs histologically appear uniform. Thus, SCLCs are currently managed as a single disease with platinum-based chemotherapy remaining the cornerstone of treatment. Recent studies have identified expression of dominant transcriptional signatures which may permit classification of SCLCs into four biologically distinct subtypes, namely, SCLC-A, SCLC-N, SCLC-P, and SCLC-I. These groups are readily detectable by immunohistochemistry and also have potential predictive utility for emerging therapies, including PARPi, immune checkpoint inhibitors, and DLL3 targeted therapies. In contrast with their histology, studies have identified that SCLCs display both inter- and intra-tumoral heterogeneity. Identification of subpopulations of cells with high expression of PLCG2 has been linked with risk of metastasis. SCLCs also display subtype switching under therapy pressure which may contribute furthermore to metastatic ability and chemoresistance. In this review, we summarize the recent developments in the understanding of the biology of SCLCs, and discuss the potential diagnostic, prognostic, and treatment opportunities the four proposed subtypes may present for the future. We also discuss the emerging evidence of tumor heterogeneity and plasticity in SCLCs which have been implicated in metastasis and acquired therapeutic resistance seen in these aggressive tumors.

## 1. Introduction

Small-cell lung cancer (SCLC) is a high-grade neuroendocrine carcinoma accounting for 15% of all lung cancers. They arise predominantly in current or former heavy smokers. Prognosis is exceptionally poor, with 5 year overall survival (OS) of 7% [1]. They are predominantly centrally located in the major airways and almost always involve the mediastinal lymph nodes. Approximately 5% of cases, however, arise peripherally in the lungs [2]. Metastasis occurs early, with up to two thirds of patients having widespread disease at initial presentation. The most common sites of metastasis include liver, bone, brain, ipsilateral and contralateral lung, and adrenal glands.

Histologically, SCLCs are homogenous and characterized by small or intermediate size cells with granular chromatin, inconspicuous nucleoli, and scant cytoplasm. High mitotic activity and apoptosis are prominent. According to the 5th edition of the WHO Thoracic Malignancies classification, most SCLCs express at least one positive neuroendocrine (NE) marker (Chromogranin A, Synaptophysin, CD56, and INSM1) on immunohistochemistry (IHC) (Figure 1) [3]. Genetically, SCLCs are homogenous also, with biallelic loss of function of TP53 and RB seen in virtually all SCLCs (Figure 2).

Currently, SCLC is treated as a single disease, with platinum-based chemotherapy remaining the cornerstone of treatment. Early-stage disease is treated with surgery and adjuvant platinum chemotherapy, locally advanced disease with concurrent radiation and platinum-based chemotherapy, and metastatic disease with systemic chemotherapy with or without immunotherapy [4]. SCLC is extremely sensitive to chemotherapy initially, with objective response rate (ORR) to first line chemotherapy of over 60% even in patients with metastatic disease [5]. Unfortunately, for the majority, this response is transient with relapse occurring early. Prognosis is poor with overall survival (OS) of less than 2 years in patients with early stage disease and approximately one year for patients with metastatic disease [1].

Over the past 20 years, there has been no improvement in the survival or response rate to chemotherapy in SCLC patients [5,6]. Therefore, there is a desperate need for a new approach in this setting. Currently, there are no biomarkers clinically available to guide treatment for targeted therapies in SCLC.

Since 2018, the PD-L1 blockade with atezolizumab and durvalumab has been incorporated as part of a frontline regimen after two large phase III trials, Impower133 [7] and CASPIAN, showed benefit [8]. However, the overall improvement in survival was modest compared to other solid tumors and only a small subset of patients seemed to derive benefit. PD-L1 IHC, a predictive biomarker for PD-L1 inhibitors in many solid tumors, failed to predict a response in SCLC and the recent approval of immunotherapy did not require PD-L1 IHC. Thus, there is a need for the development of a predictive biomarker to identify this subset of patients who can derive benefit from Immunotherapy.

Multiple recent studies have classified SCLC into four biologically distinct subtype based on differential expression of transcriptional factors: ASCL1, NEUROD1, and POU2F3 or SCLC-A, SCLC-N, and SCLC-P respectively [9,10]. A fourth subtype is negative for all three transcription factors and has a high infiltration of inflammatory cells, classified as an immune phenotype or SCLC-I [10].

There is emerging evidence that the newly defined subtypes of SCLC may have specific therapeutic vulnerabilities and therefore, may solve the current unmet need for predictive biomarkers to guide therapy selection in SCLC.

SCLC also demonstrates plasticity as well as intra- and inter-tumoral heterogeneity with implications on chemoresistance.

In this review, we present an overview of the current understanding on the biology of SCLC and how this may aid in the diagnosis, prognosis, and development of predictive biomarkers to guide treatment decisions for these aggressive tumors in the future. We also discuss the potential role of liquid biopsies in SCLC.

## 2. Pathogenesis

TP53 and RB1 (RB Transcriptional Corepressor 1) biallelic loss of function is obligatory in SCLC [11,12,13]. This inactivation of tumor-suppressor genes (TSGs) is the initiating step in oncogenesis of SCLC. This differs from most solid tumors, including non-small cell lung cancer (NSCLC), where activation of oncogenic drivers is required [14].

The vast majority of SCLCs arise in patients with a history of heavy smoking. Only rarely can they occur de novo in non-smokers [15]. This smoking etiology is reflected by the characteristic tobacco carcinogen-associated molecular signature of G > T and C > A transversions seen at high frequency (28%) and a high tumor mutational burden (TMB) [13].

Most patients are diagnosed with SCLC at an advanced stage and rarely undergo surgical resection. As a result, small tissue biopsies or cytology samples are the only biological material available. Research into the tumorigenesis of SCLC has been largely hampered by the low number of samples available. However, new research techniques such as genetically engineered mouse models (GEMMs), development of SCLC cell lines, patient-derived in vivo models, as well as analysis of mechanisms of acquired therapeutic resistance through transcriptomic and proteomic approaches, have enabled a deeper understanding of SCLC.

## 3. Genetics in SCLC

Unlike NSCLC, SCLC is a genetically homogenous tumor with concomitant inactivation of TP53 and RB1 seen in virtually all SCLC cases. Genomic studies, including whole exome and whole genome sequencing have failed to subtype SCLC, as identified mutations do not demonstrate mutual exclusivity and are consistently co-occurring [12,13,16]. In addition, no targetable oncogenic driver mutations have been identified.

TP53 is a tumor suppressor gene located on the short arm of chromosome 17, located to the most distal band of it at the position 17p13 [17]. Most mutations in SCLC are point missense mutations (63%) in the DNA binding domain site, resulting in loss of function and inability to bind the specific target gene promotors in DNA. Loss of TP53 gene function can also be a result of splice-site (15%), frameshift (13%), and nonsense (9%) mutations in SCLC.

The RB1 gene is located on the long arm of chromosome 13, at the location 13q14.2. It is a negative regulator of the cell cycle [18]. The mechanisms of RB1 inactivation in SCLC include (i) inactivating mutations, but also (ii) genomic rearrangements and translocations, resulting in a loss of gene expression. Of the inactivating mutations, most common are splice-site (31%), nonsense (30%), and frameshift (28%) mutations [13].

RB1 is usually affected by translocations, homozygous deletions, hemizygous losses, copy-neutral losses of heterozygosity (LOH), and LOH combined with polyploidy. All these events can lead to the loss of RB1 gene function in correlated gene expression studies. Thus, the RB1 gene is more often inactivated by complex genomic events which are not traceable with sequencing techniques. This might be the reason why, in earlier publications, inactivating mutations in up to 60% of RB1 gene were reported (Table 1) [19].

In rare cases of SCLC which have a preserved wild type RB1 gene, there is evidence of massive genomic rearrangements, most probably resulting from chromothripsis [13]. An example is complex massive rearrangements involving chromosomes 3 and 11, affecting the CCND1 gene, coding for cyclin-D1 on chromosome 11q13, resulting in overexpression. Cyclin-D1 is a negative regulator of the RB1 gene, resulting in RB1 gene loss of function, leading to a loss of expression measured by negative RB1 immunohistochemistry.

RB1 and TP53 are TSGs. When both alleles are inactivated, regulation of the cell cycle is lost, resulting in high proliferative activity. This genetic pattern is characteristic for all small cell carcinomas and supports the two-hit paradigm proposed by Knudson for carcinogenesis in SCLC [20]. Once this transformation is achieved, there is a 3-fold lower subclonal diversity in SCLC compared with NSCLC. In contrast to NSCLC, the level of genetic tumor heterogeneity of SCLC does not correlate with the clinical stage, highlighting differences in evolution and oncogenesis between these two lung cancers.

Amplification of MYC family genes (MYC, MYCL, and MYCN) occur in approximately 20% of SCLCs [13]. These amplifications are mutually exclusive and have been shown to play a role in the progression of SCLC [16,21]. MYC amplifications dramatically accelerate tumorigenesis and metastases of SCLC in RB1/TP53 null mice and have been implicated in SCLC plasticity [22].

Other somatic gene copy-number alterations (mutations in oncogenes) include amplification of FGFR1 and IRS2 as well as genomic losses of 3p genes FHIT (3p14) and ROBO1 (3p12) [16,23]. These alterations have been validated in DNA and RNA sequencing analysis of large cohorts of primary SCLC tumors and patient-derived and CTC-derived xenograft models [13]. Tumor suppressor genes PTEN [11] and NOTCH [24] receptors have also been functionally validated.

Recurrent mutations in epigenetic regulators including EP300, KMT2D, and CREBBP are also seen [13]. Other alterations that have been identified are depicted in Table 1.

**Table 1 cancers-14-03772-t001:** Frequently altered genes in SCLC. Data from large scale sequencing studies representing frequently altered genetic alterations in SCLC [13,25]. Data adapted from MSK-IMPACT targeted next-generation sequencing [25], MSK-MET study [26], George et al., Comprehensive genomic profiles of small cell lung cancer [13]. * RB1 mutations are likely to be underrepresented, as it is often inactivated by complex genomic events which are not traceable with sequencing techniques [13]. ** MYC family genes (MYCL1, MYCN, MYC) SCLC, Small-cell lung cancer; LOF, loss of function; GOF, gain of function; TSG, tumor suppressor gene.

Gene	Frequency in SCLC (%)	Alteration	Function	Association with SCLC Subtypes; NE Subtypes (SCLC-A, SCLC-N) Low NE Subtypes (SCLC-P, SCLC-I)
TP53	89	LOF	TSG; cell cycle regulation; transcription regulation	SCLC-A, SCLC-N, SCLC-P, SCLC-I
RB1	64 *	LOF	TSG; cell cycle regulation; transcription regulation	SCLC-A, SCLC-N, SCLC-P, SCLC-I
KMT2D	21	LOF	TSG; epigenetic regulation	-
MYC family **	19	Amplification	Oncogene; transcription regulation	SCLC-A, SCLC-N, SCLC-P
COL22A1	14	LOF	Cell-cell interaction	-
KIAA1211	13	LOF	Epithelial cell integrity	-
NOTCH1	13	LOF	TSG; cell-cell signalling	SCLC-A [27]
CREBBP	11	LOF	TSG; epigenetic regulation	SCLC-A [27]
ATRX	11	GOF	TSG; cell-cell signalling	-
FAT1	10	LOF	TSG; cell-cell signalling	-
PIK3CA	7	GOF	TSG; PTEN/mTOR signalling pathway	-
PTEN	7	LOF	TSG; PTEN/mTOR signalling pathway	-
NOTCH3	7	LOF	TSG; cell-cell signalling	SCLC-A [27]
APC	6	LOF	TSG; WNT pathway	Low-NE SCLC [28]
AIRD1A	6	LOF	TSG; epigenetic regulation	-
PTPRD	6	LOF	TSG; epigenetic regulation	-
EP300	6	LOF	TSG; epigenetic regulation	-
NF1	4	LOF	TSG; RAS signalling pathway	-
TSC2	4	LOF	TSG; PTEN/mTOR signalling pathway	-
EGFR	4	GOF	Oncogene; RAS signalling pathway	-
KRAS	3	GOF	Oncogene; RAS signalling pathway	-

## 4. Cell of Origin of SCLC

SCLC has no defined precursor lesion in humans. In contrast, carcinoid tumors, a low-grade neuroendocrine tumor of the lung, is commonly found associated with its precursor lesion, diffuse idiopathic pulmonary neuroendocrine cell hyperplasia. However, carcinoids are not smoking-associated tumors and have a completely different genetic profile than SCLC.

Originally, pulmonary neuroendocrine cells were thought to represent the cell of origin of SCLCs. The majority of SCLCs arise centrally in the large bronchi where these cells are located in their highest density. However, 5 to 15% are more peripherally located [29] where no neuroendocrine cells are found.

Multiple studies and clinical observations have now proposed any lung epithelial cell: an alveolar type 2 cell (AT2), club cell, neuroendocrine cell, or a totipotent epithelial cell have the ability to form SCLC [30]. This is supported by SCLC arising in acquired tyrosine kinase inhibitor (TKI)-resistant EGFR mutated lung adenocarcinomas. Transformed tumors maintain the EGFR mutation, in addition to RB1 loss, suggesting direct evolution from the original tumor rather than a distinct primary cancer [31]. In contrast, resistant NSCLCs which do not develop an SCLC component rarely demonstrate RB1 loss [32]. This finding suggests that (a) SCLC can develop from the same cell as lung adenocarcinoma (AT2 cells), and (b) RB1 loss is a key step in the pathogenesis of SCLC.

Multiple animal studies have also identified that SCLC can be induced in multiple pulmonary epithelial cell types [33]. Chen et al. specifically targeted different pulmonary epithelial cells in GEMMs to have a combined gain of MYC expression with TP53/RB1 deletion. Cells targeted included AT2 cells, club cells, and neuroendocrine cells. These genetic changes exclusively resulted in SCLC with 100% penetrance irrespective of cell lineage [34]. This study also identified that the cell of origin determined differences in the size and anatomical location of tumor masses: SCLCs originating from neuroendocrine cells lead to a small number of large invasive tumors arising from the large bronchi, while tumors originating from AT2 cells and club cells developed numerous small tumors in distal bronchioles and alveolar spaces, reflecting the cell location and distribution in normal lung [35,36].

The cell of origin is also thought to also initiate metastasis through different molecular mechanisms [37]. NFIB upregulation represents a key molecular mechanism of metastasis in a subset of SCLC, and can occur early during tumorigenesis [38]. Mechanisms of NFIB-driven metastasis remain unclear but may have a pro-metastatic role by inducing gene expression programs involved in cell adhesion, migration, and neuronal differentiation [39].

Yang et al. induced SCLC in RPR2 (Rb1^fl/fl^Trp53^fl/fl^Rbl2^fl/fl^) mice through transduction of epithelial cells with Adenoviral-CMV-Cre [37]. Metastasis in this model occurred through NFIB upregulation. The transduced cells included several pulmonary epithelial cells, and therefore, the cell type that gave rise to this SCLC is still unknown. However, when SCLC is initiated in the same RPR2 models specifically from pulmonary neuroendocrine cells, the resulting metastasis does not upregulate NFIB [37]. This highlights that the cell of origin might be important in metastatic potential in SCLC.

Additionally, the cell of origin may determine distinct SCLC subtypes. For example, SCLC-P, an emerging subtype with a low-NE phenotype and the presence of MYC amplification, is thought to arise from tuft cells [40] (discussed later). Multiple studies have identified these subtypes as having distinct therapeutic vulnerabilities [9,10,41].

Together, these recent findings suggest that genetic changes, rather than cell of origin, leads to the development of SCLC. However, the cell of origin may determine tumor location, disease latency, metastatic potential, and therapeutic vulnerabilities rather than genetic drivers. To what extent the cell of origin is correlated to the biology of SCLC is still an unsolved yet intriguing question.

## 5. Emerging Subtypes in SCLC

Although SCLC is histologically and genetically homogenous, multiple studies have consistently shown biological heterogeneity based on the expression of specific transcription factors [9,10]. So far, neither clinical nor genetic features can distinguish SCLC.

Taking several lines of evidence, from SCLC primary human tumors, patient-derived xenografts, cancer cell lines, and genetically engineered models, Rudin et al. proposed four major subtypes of SCLC [9]. These subtypes are defined by a high expression of four key transcription regulators, three of which are achaete-scute homologue 1 (ASCL1), neurogenic differentiation factor 1 (NEUROD1), POU class 2 homeobox 3 (POU2F3) [9], and they are referred to as SCLC-A, SCLC-N, SCLC-P, respectively. The fourth subtype, the yes-associated protein 1 (YAP1), referred to as SCLC-Y, was found in earlier gene expression studies to be associated with poor prognosis, shorter patient survival and increased chemoresistance [42]. The validation of SCLC-Y as a distinct subtype, however, failed in multiple recent studies using IHC and single cell sequencing [10,43]. High YAP1 expression may have originated from an admixed NSCLC component in combined SCLCs in these early studies [43]. Instead, Gay et al. proposed a fourth subtype which has no known dominant transcriptional regulator [10]. This group expresses epithelial to mesenchymal transition (EMT) markers and has high expression of genes related to immune cell infiltration, immune checkpoints, HLAs, and INF-y activation, and it is therefore named the ‘inflamed’ subtype, or SCLC-I. This study also further validated SCLC-A, SCLC-N, and SCLC-P as distinct subtypes. SCLC-A and SCLC-N represent neuroendocrine (NE) subtypes while SCLC-P and the fourth “triple negative” SCLC-I group represent low-NE subtypes.

Gay et al. identified these four subtypes using unbiased clustering of RNA-sequencing data from 81 patients with limited stage and surgically resected SCLC, and 276 treatment naïve patients from the Impower133 study with extensive stage SCLC. These four subtypes were then validated in cell lines and tumor samples from treatment-naïve metastatic patients, indicating that these subtypes are neither stage- nor treatment-specific [10].

### 5.1. Neuroendocrine Subtypes

Both SCLC-A and SCLC-N exhibit similar immunophenotypic and pathological features compared to other subtypes. They are positive for NE markers synaptophysin, CD56, chromogranin A, and INSM1. Furthermore, they show expression of TTF-1 and DLL-3. Thus, these groups are defined as NE groups (NE^+^/TTF-1^+^/DLL-3^+^).

ASCL1 and NEUROD1 are both basic helix–loop–helix transcription factors that activate neuroendocrine genes and are key determinants of differentiation and maturation of pulmonary neuroendocrine cells (PNECs) in the developing lung [44,45].

ASCL1 is essential for the development of PNECs and is expressed in these cells in the adult lung [46]. Deletion of ASCL1 in TP53/RB1 knockout mice abolishes the development of SCLC, confirming that ASCL1 is necessary for the initiation and development of at least a subset of SCLC [46]. As well as activating neuroendocrine differentiation, ASCL1 regulates stemness, cell cycle progression, and mitosis, maintaining tumor development and survival [46,47]. SCLC-A is the most common subtype, with 70% of SCLCs expressing ASCL1 on the RNA [9] and protein level (IHC) [43].

Inhibition of the Notch pathway is an important mechanism to maintain NE phenotype and carcinogenesis in SCLC [48]. Delta-like ligand 3 (DLL3) is a Notch receptor ligand that results in inhibition of the Notch pathway, therefore promoting NE differentiation [49]. Notch and ASCL1 are mutually exclusive, with Notch acting as a transcriptional repressor of ASCL1 expression [50]. DLL3 is a direct downstream target of ASCL1, which interacts with the DLL3 gene promoter, ensuring that the Notch pathway is inhibited. In normal cells, DLL3 is located in the Golgi and in cytoplasmic vesicles and is negative on the cell plasma membrane [51]. In contrast, in NE SCLCs (SCLC-A and SCLC-N), it is overexpressed on the cell membrane and is intracytoplasmatically about 35-fold compared to the normal lung [51]. In contrast, DLL3 expression is entirely negative in low-NE subtypes at the protein and genetic level [43,52].

MYCL regulates neuronal developmental pathways and an NE phenotype and is highly expressed in SCLC-A. In contrast, MYC is incompatible with SCLC-A and leads to Notch signalling and EMT. MYC also drives the transdifferentiation from SCLC-A to SCLC-N and then to a low-NE SCLC in a stepwise fashion [53,54].

SCLC with NEUROD1 expression (SCLC-N) also exhibits an NE phenotype. However, unlike ASCL1, deletion of NEUROD1 in TP53/RB1 knockout mice has no effect on the initiation or progression of SCLC, and therefore, seems not to be required for its development [46]. Instead, SCLC-A may switch to SCLC- N in a stepwise fashion driven by MYC [54]. Further supporting this theory, SCLC-A has been shown to be significantly overrepresented in primary tumors. In contrast, SCLC-N is enriched in nodal and distant metastases [55]. The expression of NEUROD1 is 11% on the RNA [9] but 45% on the protein level (IHC) [43]. This difference is thought to be due to bulk gene expression not being sensitive at identifying weakly positive cells. These studies also included different patient populations. Therefore, validation and correlation of NEUROD1 IHC and RNA data in the same patients may resolve this issue. In addition, multiple studies have identified that up to 37% of SCLCs have subpopulations of cells that co-express both ASCL1 and NEUROD1, suggesting these two might be closely related or even may be in a transition state [10,43]. This interpretation is further supported by the very similar survival and therapy response of these two subtypes in the Impower133 trial (randomization for immunotherapy in addition to the standard chemotherapy) [10].

In summary, SCLC-A and SCLC-N, both NE-high groups, may represent truly distinct biological subtypes or may represent different steps in the evolution of NE-high SCLC. However, further studies are needed to confirm this.

### 5.2. Low-Neuroendocrine Subtypes

ASCL1^-^/NEUROD1^-^ SCLCs represent low-NE subtypes (NE^low/-^/TTF-1^low/-^/DLL-3^low/-).^ These subtypes are either uniquely associated with POU2F3 (SCLC-P), or lack a known dominant regulator [10,43].

POU2F3 is a transcription factor that binds to a specific octamer DNA motif to regulate cell type-specific differentiation and is a master regulator of tuft cells [40]. Tuft cells are chemosensory cells within lung parenchyma which respond to external stimuli by releasing bioactive substances to regulate epithelial and immune cell functions, similar to NE cells [56]. SCLC-P tend to locate where tuft cells lie (primary and secondary bronchi) and demonstrate similar expression profiles [40]. Tuft cells have therefore been hypothesized as the cell of origin for this subtype of SCLC. It is important to note that tuft cells lack the expression of usual NE markers and therefore, SCLC-P tend to be NE^low/-^ [57].

POU2F3 expression is observed in 7% of SCLCs by IHC [43] and represents a disease entity distinct from the classical NE form of disease, virtually never co-expressing with ACSLC1 or NEUROD1 [40]. Although tuft cells are hypothesized to be the cell of origin, POU2F3-expressing SCLCs have been shown to be enriched in combined SCLCs compared to ASCL1/NEUROD1 subtypes [43,58]. This suggests that either (a) SCLC-P and NSCLC share a common cell of origin, or (b) SCLC-P demonstrates greater plasticity than ASCL1/NEUROD1 subtypes. As the SCLC-P subtype is truly distinct from other SCLCs this may potentially impact tumor biology, including progression, prognosis, and metastatic ability. However, further studies are needed to prove this.

The fourth subtype is a low-NE group that lacks a known dominant transcription regulator. Gay et al. identified this fourth subtype as having the highest expression of genes related to immune cell infiltration, immune checkpoints, HLAs, and INF-y activation, and has the highest levels of immune cell infiltrate, including NK cells, T cells, and macrophages [10]. Therefore, this subtype was termed the ‘inflamed’ subtype (SCLC-I). This subtype also has the most mesenchymal phenotype, expressing vimentin, AXL, and very low levels of epithelial markers, including E-Cadherin (CDH1). This subtype has been shown to be implicated in SCLC plasticity and chemoresistance, with SCLC-A switching to SCLC-I after first line chemotherapy [10]. After this switch occurs, chemoresistance develops, most probably as a consequence of EMT transition. This subtype showed the best responses to immunotherapy in the Impower 133 trial (discussed later) [10].

### 5.3. SCLC with YAP1 Expression

YAP1 is a transcription regulator activated by the HIPPO growth signaling pathway [59] and it has been shown to be expressed in 2 to 10% of SCLCs [9,54]. YAP1 expression is enriched in low NE SCLCs and in SCLCs that express multiple EMT-associated markers, including Vimentin, SNAI2, and CD44 [54]. Multiple studies, however, have failed to distinguish YAP1 expression as a distinct subtype [10,43]. Baine et al. identified YAP1 to be expressed in 33% of SCLCs by IHC [43] but this expression was at very low levels and was found across subtypes, albeit at higher frequencies in low NE groups. Although YAP1 does not define a distinct subtype, its expression is thought to represent reprogramming of SCLC from an NE-high to NE-low phenotype. This reprogramming is driven by MYC activation of the Notch signaling pathway [54]. Additionally, SCLCs with high YAP1 expression display multidrug resistance in both in vitro and in vivo assays [60]. Therefore, although not a distinct subtype, YAP1 expression may detect plasticity within SCLCs switching from an NE to a low-NE state, EMT, and development of chemoresistance [60].

## 6. SCLC Heterogeneity and Plasticity

Several studies have shown that SCLC can be subdivided into groups dependent on master transcriptional activators, with IHC analysis revealing that tumors largely represent single subtypes. However, some tumors have small populations of cells expressing two or more transcription activators [10,43], as emerged largely from single cell studies. SCLC-A, SCLC-N, and SCLC-I have been shown to exist within a single tumor [10,54] with higher heterogeneity seen following platinum resistance [55].

Additionally, SCLC has the ability to switch subtype from NE to low-NE subtypes [54]. SCLC-P appears to be truly distinct, with co-expression rarely seen with other subtypes. Additionally, a subtype switching to SCLC-P has not been demonstrated in any studies to date.

This intratumoral plasticity and heterogeneity seen in SCLC has been shown to enable tumor cells to grow, metastasize, and to develop chemoresistance [55]. How SCLCs develop plasticity requires further investigation, although a few mechanisms have been proposed.

### 6.1. Notch and MYC Signaling Pathways

As mentioned above, Notch inhibition is important to maintain the NE phenotype with DLL3, a Notch ligand inhibitor, highly expressed in SCLC-A and SCLC-N.

Ireland et al., using mouse and human models with time-series single transcriptome analysis, found that MYC directly activates Notch signaling to reprogram SCLCs from an NE high to an NE low state (Figure 3). This was done in a stepwise fashion, with subtypes switching from SCLC-A to SCLC-N (both NE subtypes) to a low-NE state expressing YAP1 [54]. This MYC-driven transdifferentiation increases following chemotherapy and during acquired resistance (Figure 3) [10,28,54,55]. Stewart et al. reported that SCLC-A is significantly overrepresented in primary tumors. In contrast, SCLC-N is enriched in nodal and distant metastases [55], suggesting that plasticity of SCLC driven by MYC may enable metastatic ability. However, due to the paucity of tissue samples in SCLC, it is not clear if metastasis tend to exhibit NE or low NE states.

### 6.2. Plasticity and Chemoresistance

Intratumoral heterogeneity (ITH) and plasticity have been identified as a mechanism for gaining metastatic ability and developing chemoresistance frequently seen in SCLC. Gay et al. revealed that ITH is a dynamic process with transcriptional shifts away from SCLC-A subtype to SCLC-I during platinum treatment [10]. In this study, two SCLC-A treatment-naïve platinum-sensitive CDX models were treated with cisplatin to a maximal response. Single-cell RNA sequencing (scRNAseq) of these treated models identified a reduction in the proportion of SCLC-A cells, and an increase in the SCLC-I proportion (Figure 3). These cells were platinum-resistant and had higher EMT scores, suggesting that subtype switching could underlie platinum resistance.

Stewart et al. developed platinum-sensitive CDX models from SCLC patients and subjected them to extensive treatment with cisplatin to develop relapse [55]. Again, using scRNAseq, untreated tumors were molecularly homogenous, while relapsed tumors were associated with increased ITH, with distinct cell populations expressing EMT genes (Figure 3) [55]. These studies highlight an evolving transcriptional complexity during treatment which may exert profound effects on therapeutic responses. The window for therapeutic vulnerability, therefore, is short in SCLC and aggressive frontline strategies may be the most appropriate treatment strategy. Alternatively, therapeutic switching should occur throughout a patient’s management to adapt to the changing landscape of SCLC.

### 6.3. SCLC-P and Plasticity

SCLC-Ps represent 7% of SCLCs and are thought to represent a truly distinct subtype. Due to the paucity of tissue samples and lack of animal models, most studies fail to have adequate representation of this subtype, making assessment/biology difficult to study. SCLC-P subtype has a high expression of MYC [40]. However, POU2F3 mRNA expression is observed at extremely low counts during the MYC-driven transition from an NE to a low-NE state as described above [54]. This observation suggests that MYC is not causal for the SCLC-P subtype and that distinct pathways/cell of origin might be driving this subtype.

In a recent study, Caser et al. obtained a set of POU2F3-driven PDXs from a male patient in his 70s with extensive SCLC [58]. The initial biopsy revealed combined SCLC and adenocarcinoma. Ten biopsies were obtained from metastatic sites including lung, chest, liver, diaphragm, and rib (Figure 3). All biopsies were POU2F3-positive and had MYC amplification by IHC and RNA-sequencing, respectively. ASCL1, NEUROD1, and YAP1 were not expressed. There were shared somatic mutations as well as POU2F3 in the adenocarcinoma and SCLC, suggesting that POU2F3 can derive from the same cell as lung adenocarcinoma. POU2F3 was the only subtype and was present in all biopsies, suggesting SCLC-P truly is distinct from other subtypes and may demonstrate less plasticity. Baine et al. also identified an association with POU2F3 expression and combined SCLC suggesting a closer relationship with NSCLC than the other subtypes [43].

### 6.4. PLGC2 and Intratumoral Heterogeneity

Phospholipase C Gamma 2 (PLCG2) expression in a subpopulation of cells has been shown to confer the aggressiveness seen in SCLC [61]. Chan et al. sequenced 155,098 transcriptomes from 21 human biospecimens, including 54,523 SCLC transcriptomes to identify heterogeneity within SCLC [61]. Unbiased clustering identified a subpopulation of cells with a high expression of PLGC2 demonstrating a malignant phenotype, enriched for genes related to metastasis, chemotaxis, and stemness, including genes involved in WNT and BMT signaling and EMT markers. This PLGC2 cluster was recurrent in SCLCs and was not limited to any subtype.

PLCG2 is a transmembrane signaling enzyme that is involved in transmitting signals from growth factor receptors and immune system receptors across the cell membrane [62]. PLCG2 has been shown to be involved in tumor microenvironment (TME) remodeling and has been identified as an immune-related gene in soft tissue sarcomas and colon cancer [63,64].

The relative proportion of this subpopulation in each tumor had a significant impact on prognosis. SCLCs with >0.75% representation of this subpopulation of cells were associated with a significantly reduced overall survival (OS). Taken together, these data support that a subpopulation of stem-like, pro-metastatic cells with high PLCG2 expression have a remarkable prognostic impact. This cluster/population of cells was identified across subtypes, treatment groups, and tissue locations, pointing to a potentially universal characteristic of SCLC.

In this study, there was only one SCLC-P and therefore, the study focused on SCLC-A and SCLC-N. Again, SCLC-P being underrepresented due to a paucity of samples.

## 7. Potential Clinical Utility of Emerging Subtypes

The detection of biological heterogeneity based on master transcriptional activators has been a huge advancement in our understanding of SCLC and has potential to translate into clinical practice in the future. Each of these subtypes have distinct biological activity and therapeutic vulnerabilities which will hopefully lead to the development of novel diagnostic, prognostic, and predictive biomarkers to aid in the pathologic assessment and guide clinical decisions for these aggressive tumors.

### 7.1. Subtypes as Emerging Diagnostic Biomarkers

Diagnosis of SCLC is made by histopathological examination. Because of the central location of the tumors, biopsies are most often obtained by bronchoscopy with or without endobronchial ultrasound. As a result, diagnosis is made on scant biopsy specimens or cytology. Due to the aggressive nature of SCLC, workup including diagnosis and staging should be performed as quickly as possible after presentation.

Morphologically, SCLCs are composed of small cells (<3 resting lymphocytes) with scant cytoplasm, finely granular nuclei, and inconspicuous nucleoli, indistinct cell boarders, high mitotic count, apoptosis, and frequent necrosis. Tumors are usually densely packed, forming a sheet-like pattern [3]. Morphologically, SCLC are homogenous compared to NSCLC, although rare variations exist [65].

Historically, SCLC is a morphological diagnosis made on the haematoxylin and eosin (H&E) stain. However, IHC is widely used in pathology practice. Histologically, the differential diagnosis includes other neuroendocrine tumors, especially large cell neuroendocrine carcinoma (LCNEC), basaloid squamous cell carcinoma, undifferentiated SMARCA4 tumor, small round blue cell tumors, and lymphoma. Both clinical course and treatments vary widely across these malignancies, and therefore, accurate diagnosis is essential.

NE markers are positive in 90 to 95% of SCLC cases. However, 5–10% can be positive only with CD56, sometimes only weakly, or even negative, creating a major diagnostic challenge especially in those where morphology is not well preserved.

As a result, large IHC panels are needed to exclude other differentials, often delaying diagnosis. In such cases, or whenever the differential diagnosis with low-grade NE tumors, NSCLC or LCNEC, arises, the loss of RB IHC and a mutant staining for TP53 (either loss of TP53 or a strong diffuse staining) is helpful. This pattern is seen in SCLC representing its genetic hallmark (Figure 2) [13]. This approach is not yet validated but might be of great value in small or crushed biopsies. In difficult cases, the addition of p16 IHC might be of additional assistance; strong and diffuse p16 reflects loss of Rb through its negative regulatory loop [66].

POU2F3, the novel transcriptional factor expressed in 7% of SCLC, is showing great promise as a diagnostic biomarker in NE^low/-^SCLCs. Baine et al. analyzed POU2F3 IHC expression in SCLC (n = 123) and other major lung cancer types (n = 433) [57]. Expression was diffusely positive in 70% of SCLCs that were fully NE negative. Expression was negative in NE SCLCs and therefore, is best suited as a second step marker in suspected NE^low/-^SCLC. Although not fully restricted to SCLC (positive staining was seen in basaloid SCC and LCNEC in 22% and 12% of cases, respectively), it is highly selective, and therefore is useful as a multipurpose IHC, similar to TTF-1. The addition of POU2F3 IHC, therefore, would aid in a more efficient diagnostic pathway for NE^low/-^SCLCs, eliminating the use of larger IHC panels and conserving tissue for a further workup, such as detection of predictive biomarkers, in the future.

### 7.2. SCLC Subtypes as Prognostic Biomarkers

None of the four distinct subtypes give clear prognostic information. SCLC-A and SCLC-N are associated with immune cold tumors [59] and low NE subtypes are associated with chemoresistance. However, few studies have evaluated the prognosis of these subtypes. SCLC-P has been identified as a poor prognostic factor in one study [67] and conflicting results exist as to whether high YAP1 expression is associated with poor prognosis [42,59]. For now, SCLC subtype behavior in a clinical context is still not clarified.

### 7.3. Predictive Biomarkers

In stark contrast to NSCLC, SCLC has been treated as a single disease, with a one-size-fits-all model, and with platinum-based chemotherapy remaining the cornerstone of treatment. Over 20 years, there has been no improvement in the survival or response rate to chemotherapy of SCLC patients and therefore, there is a desperate need for new approaches in this setting [5].

Currently, there are a number of promising biomarkers in the pipeline which may be used to guide clinical decisions for these patients in the future.

#### 7.3.1. SCLC Subtypes as Predictive Biomarkers

The four described SCLC subtypes, SCLC-A, SCLC-N, SCLC-P, and SCLC-I, described by distinct translational activators, have been shown to have distinct therapeutic vulnerabilities. Using in vitro cytotoxic assays, Gay et al. identified that the SCLC-A subtype have increased sensitivity to BCL2 inhibitors, the SCLC-N subtype, to aurora kinase inhibitors, and the SCLC-P subtype, to PARPi and antimetabolites (Figure 4) [10]. The SCLC-P subtype had increased sensitivity to PARPi independent of SLFN11 (an emerging predictive biomarker for PARPi in SCLC, see below), suggesting there is an alternate pathway of efficacy which potentially widens the cohort of patients who will derive benefit from this promising new target in SCLC. Within SCLC-A, there is a bimodal expression of SLFN11, and if separated by high and low expression, there is a stark difference in cisplatin sensitivity and to PARPi with olaparib [10].

To identify therapeutic vulnerabilities, Gay et al. also identified unique protein expression patterns between groups which may serve as predictive biomarkers. SCLC-A and SCLC-N subtypes have a high expression of DLL-3, which is absent in SCLC-P and SCLC-I groups. A subset of SCLC-A also has high expression of SLFN11, which predicts a response to PARPi (see below). The SCLC-N group has a high expression of somatostatin receptor 2 (SSTR2), which can be targeted by somatostatin analogues such as octreotide.

The response of subtypes to these targeted therapies requires validation. However, the identification of distinct therapeutic vulnerabilities within these four subtypes highlights that a one-size-fits-all-model may not be appropriate for managing this disease. This study will certainly pave the way for biomarker-driven clinical trials in the future introducing SCLC into the world of precision oncology.

#### 7.3.2. SCLC-I Predicts Response to Immunotherapy

SCLCs have a high TMB and, therefore, are predicted to induce strong T-cell responses. NSCLC are also associated with a similarly high TMB, and as such, have shown responsiveness to immunotherapy. Immunotherapy by means of targeting PD1/PD-L1, either as a monotherapy or in combination with chemotherapy, has become the first-line treatment for patients with stage IV NSCLC lacking a driver mutation [68]. PD-L1 expression on tumor cells (tumor proportion score) by IHC is a strong predictor of response, with cut offs of ≥1% and ≥50% used as selection criteria for combination therapy and monotherapy, respectively [69,70,71]. Additionally, TMB status is an important predictive biomarker with high TMB predicting improved objective response to durable benefit, and progression-free survival independent of PD-L1 expression [72]. High TMB in NSCLC is associated with immune cell infiltration and an inflammatory T-cell mediated response which may explain the increased sensitivity to immunotherapy [73].

Although NSCLC and SCLC have a similar TMB profile, they exhibit a differential response to immunotherapy. Clinical trials involving immunotherapy in SCLC have yielded disappointing results, with only a subset of patients deriving a benefit [74]. PD-L1 IHC is rarely expressed in SCLC and does not predict a response to immunotherapy [75]. The differential responsiveness between SCLC and NSCLC may lie within the interaction of SCLC and its surrounding environment, including immune surveillance [76]. SCLC exhibits an extremely cold T-cell receptor (TCR) repertoire and lower immune infiltrate compared to NSCLCs despite having similar TMBs [77]. SCLCs also have low infiltration by immune cells, specifically, cytotoxic T cells [78].

Currently, there are no biomarkers that predict a response to immunotherapy in SCLC.

The novel, low-NE subtype identified by Gay et al., SCLC-I, demonstrates an ‘inflamed’ phenotype and has a high expression of both CD8A and CD8B, suggesting greater cytotoxic T-cell infiltration [10].

The Impower133 trial was the first randomized trial demonstrating PFS and OS improvements with Immunotherapy in SCLC [75]. Using data from this trial, Gay et al. retrospectively stratified survival data by SCLC subtype and the identified OS benefit was seen in the SCLC-I group compared to all others in the platinum-etoposide (EP) plus atezolizumab arm, but not in the EP plus placebo arm (HR, 0.566; 95% CI, 0.321–0.998) [10]. This represents clinical data that the SCLC-I subtype is a candidate biomarker for predicting a response from the immune checkpoint blockade in SCLC.

Additionally, the SCLC-N subtype has been shown to have the most immunosuppressive TME and therefore, may derive no benefit from IO [61]. These studies highlight the need to match tumor subtype to therapy to maximizes the response in patients. Future prospective trials are needed to elucidate the clinical relevance of the novel transcriptional subsets in SCLC.

#### 7.3.3. SLFN11 as a Predictive Biomarker

Resistance to cell death and genomic instability are important hallmarks of cancer [79]. Small molecule inhibitors that directly damage DNA response (DDR) have gained increasing interest over the years. Poly (ADP-ribose) polymerase (PARP) are a family of proteins who play a critical role in DNA repair. When induced by a DNA lesion, PARP catalyzes the addition of poly (ADP-ribose) chains to target proteins, which mediates the recruitment of additional DNA repair factors to the damaged DNA [80]. PARP inhibitors (PARPi) have been studied most in cancers with deficient homologous recombination repair (HRR) pathways which depend on PARP-mediated base excision for repair.

Schlafen 11 (SLFN11) is a DNA/RNA helicase that induces an irreversible replication block and it has been identified as a predictive biomarker of DNA-damaging agents and PARPi in preclinical settings in SCLC [81], as well as many other solid tumors, including breast [82], ovarian [83], colorectal [84], and gastric cancer [41], and mesothelioma [85].

SLFN11 also predicts the response to other DNA-damaging agents, including platinum salts, topoisomerase I inhibitors, topoisomerase II inhibitors, alkylating agents, and antimetabolites. Low expression also predicts resistance to these agents [86].

Clinical detection of SLFN11 can be made using IHC. Multiple studies have demonstrated that detection of SLFN11 is made by the presence of any nuclear staining [86]. Conversely, negative SLFN11 tumors have a complete absence of staining. Therefore, the distinction between SLFN11-positive and -negative tumors can be easily made in clinical practice by IHC.

Unlike other biomarkers, SLFN11 induces its cellular response to PARPi independently of HRR. Instead, ‘PARP trapping’ occurs, whereby PARPi traps PARP1/2 at the site of DNA damage preventing repair [87]. SLFN11 induces cell cycle arrest in the S-phase, which leads to a stalling of replication forks resulting in fork breakage and replisome disassembly [88]. SLFN11 and PARPi are believed to have synergistic effects through this ‘trapping’ mechanism [89]. PARP trapping varies among different PARPi, with talazoparib being the strongest and veliparib being the weakest [86,87,89,90,91]. The predictive strength of SLFN11, therefore, may be PARPi class dependent.

A phase II trial of relapsed SCLC treated with temolizomide plus veliparib (PARPi) versus temolizomide plus placebo found no significant difference in 4-month PFS or median OS [92]. Retrospective analysis, identifying SLFN11-positive patients using IHC on archival tissue, identified that SLFN11-positive patients had a significantly longer PFS and OS compared to SLFN11-negative patients in the temozolomide plus veliparib group [92]. This study represented the first clinical data that SLFN11 is a candidate biomarker for PARPi in SCLC.

A phase II randomized clinical trial of atezolizumab in combination with talazoparib versus atezolizumab alone in SCLC is currently ongoing (NCT04334941) for patients with SLFN11-positive extensive-stage SCLC. As mentioned above, talazoparib has the strongest PARP-trapping ability and may confer additional benefit compared to other PARPi in SLFN11-positive patients. This is the first clinical trial selecting patients based on SLFN11 status.

Lurbinectedin, an alkylating agent, recently received FDA-accelerated approval as a second line treatment option for metastatic SCLC [93]. SLFN11 has been shown as a candidate biomarker for lurbinectedin sensitivity. SLFN11-low SCLC cell lines are resistant to lurbinectedin [94]. However, the addition of an ATR (Ataxia telangiectasia mutated and Rad3-related) inhibitor re-sensitizes these cells [94]. This confirms SLFN11’s role as a master regulator of DNA damage response, independent of ATR, with the absence of SLFN11 leading to synthetic lethality when ATR is inhibited. Therefore, the addition of an ATR inhibitor to lurbinectedin in SLFN11-low tumors may overcome resistance.

SLFN11 has also been shown to be detectable in CTCs in SCLC and may represent a candidate biomarker to monitor patients via non-invasive liquid biopsy [95] (discussed below).

#### 7.3.4. DLL3 as a Predictive Biomarker

Delta-like ligand 3 (DLL3) is a ligand that inhibits the Notch pathway and is upregulated and aberrantly expressed on the cell surface in up to 80% of SCLCs, specifically, NE subtypes (SCLC-A and SCLC-N) [52,96]. In contrast, DLL3 is only expressed at very low levels in a few normal cells and is exclusively cytoplasmic [51]. This selective expression has gained a lot of attention as an attractive biomarker in SCLC.

Rovalpituzumab tesirine (Rova-T) is an antibody–drug conjugate composed of a DLL3-targeting immunoglobulin G1 monoclonal antibody tethered to pyrrolobenzodiazepine (PBD), a toxic DNA crosslinking agent, by means of a protease-cleavable linker [52]. Rova-T was designed to deliver cytotoxic treatment to DLL3, expressing SCLC cells while sparing healthy cells, reducing side effects. Phase II and III clinical trials using RovaT as a third line and beyond treatment for participants with relapsed or refractory DLL3 expressing SCLC exhibited an inferior OS compared to patients on current standard second-line chemotherapy, topotecan, and therefore, it was discontinued [97].

Despite this, DLL3 remains an attractive target due to its specificity and high expression in SCLC. Novel therapeutic targets, such as anti-DLL3 bispecific T-cell engager, are in development [98]. Additionally, a recent study used a radiolabeled anti-DLL3 mAb SC16 with the therapeutic radioisotope Lutetium-177 in GEMMs. This radioisotope specifically attaches to SCLC cells to deliver targeted radiotherapy, minimizing radiation to healthy cells. Results showed antitumor efficacy and a low toxicity profile, making it a strong potential for clinical translation [99].

## 8. Potential Use of Liquid Biopsies in SCLC

Due to the scarcity of tissue and the dynamic nature of SCLC demonstrating plasticity, liquid biopsies represent an attractive alternative for identifying biomarkers and for longitudinal disease monitoring.

Reflecting their high metastatic rate, SCLCs have one of the highest concentration of CTCs among solid tumors, being detected in up to 85% of patients [100]. This high concentration in combination with improved technologies has made the ability to detect CTCs clinically feasible [101]. CTCs in SCLC have many potential roles. CTCs are highest at time of diagnosis and relapse and are suppressed while on treatment [100], and therefore, they are an attractive tool to monitor disease status. Baseline CTCs are also prognostic with CTCs >50/7.5ml of blood having inferior OS [102]. CTC concentration does not correlate with tumor size and stage [103] and may reflect the biological diversity of SCLCs.

Another use for CTCs is to evaluate cellular changes as a surrogate for the whole tumor via biomarker detection. SLFN11, a promising predictive biomarker for several therapeutics including platinum, PARPi, topotecan, and lurbinectedin, has been shown to change over time in preclinical studies [81]. Zhang et al. developed an immunofluorescence assay to identify SLFN11 in CTCs and found a comparable proportion of SLFN11-positive CTCs to that in tissues samples [95]. Interestingly, CTCs showed changes in SLFN11 expression during therapy: downregulation as early as 72 hrs following treatment with platinum-based chemotherapy [81]. Conversely, CTC concentration persists in patients with clinical progression while SLF11 expression remains downregulated. This study reflects the dynamic nature of SLFN11 during treatment. Therefore, SLFN11 biomarker status on archival tissue may not represent real-time expression in patients. This study also highlights the clinical feasibility of biomarker detection via liquid biopsies in these patients.

Due to the difficulty in obtaining adequate tissue, as well as the functional plasticity of SCLCs, longitudinally monitoring patients through liquid biopsies (identifying CTCs or biomarker detection) should be considered an important tool for guiding treatment in SCLC.

## 9. Conclusions

Recent studies have led to a deeper understanding of the biological diversity of SCLC. SCLC can be subclassified based on the expression of different transcriptional factors. The four subtypes SCLC-A, SCLC-N, SCLC-P, and SCLC-I, have distinct therapeutic vulnerabilities and demonstrate different plasticity and intratumoral heterogeneity when exposed to platinum-based chemotherapies. Additionally, expression of DLL3 and SLFN11 may have a significant prognostic and predictive value. Including the novel subtypes into the routine diagnostics of SCLC is relatively easy using IHC. Although there are still many unanswered questions on the biology of SCLC, novel classification might pave the way for novel therapeutic possibilities for these different subtypes in the future.

## Figures and Tables

**Figure 1 cancers-14-03772-f001:**
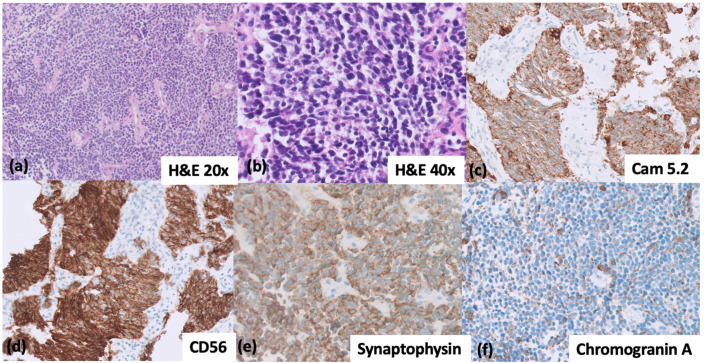
(**a**–**f**) Immunohistochemical patterns for the diagnosis of small-cell lung cancer. (**a**,**b**) H&E image of SCLC at 20× and 40× magnification, respectively. (**c**) CAM 5.2 with a dot-like pattern of cytoplasmic positivity; (**d**) CD56, strong positivity; (**e**) Synaptophysin, diffusely positive; (**f**) Chromogranin A, dot-like positivity in cytoplasm, pathognomonic of SCLC. (**b**–**f**) taken at 40× magnification.

**Figure 2 cancers-14-03772-f002:**
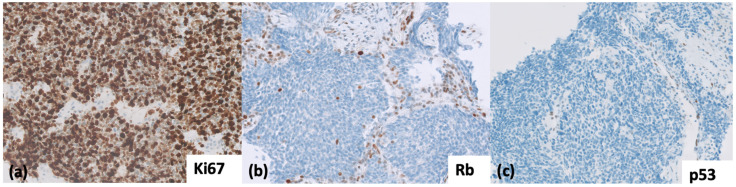
(**a**) Ki-67, demonstrating a high proliferation index > 80%, commonly seen in SCLC (40× magnification). (**b**) RB, loss of nuclear staining in the tumor nuclei; note the wild type pattern of staining in the background stromal cells (40× magnification); (**c**) p53 IHC with null mutation pattern; note, again, the wild type staining in the background capillaries and stromal cells (20× magnification).

**Figure 3 cancers-14-03772-f003:**
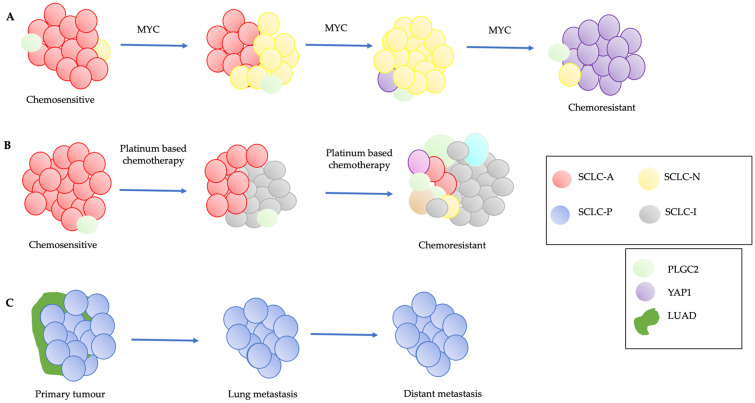
(**A**–**C**). Representation of plasticity and intratumoral heterogeneity leading to chemoresistance and metastasis in SCLC subtypes. (**A**) MYC drives the temporal evolution of SCLC-A to SCLC-N to SCLC expressing YAP1 [54]. The former two are NE subtypes and the latter, a low NE subtype. SCLCs expressing YAP1 have been shown to demonstrate multidrug resistance [60]; (**B**) Acquired resistance to platinum-based chemotherapies leads to a decrease in the proportion of SCLC-A cells, and an increase in the SCLC-I proportion [10]. SCLC-I cells are platinum-resistant and have higher EMT scores, suggesting that subtype switching could underlie platinum resistance. SCLCs with acquired resistance to platinum-based chemotherapies are molecularly heterogeneous compared to their treatment-naïve counterparts [55]; (**C**) SCLC-P is a distinct subtype of SCLC rarely co-expressing other subtypes or YAP1. An association between SCLC-P and combined SCLC (with lung adenocarcinoma) has been made [43,58]. SCLC-P is a low NE subtype that is retained in primary and metastatic tumors [58]. (LUAD; lung adenocarcinoma).

**Figure 4 cancers-14-03772-f004:**
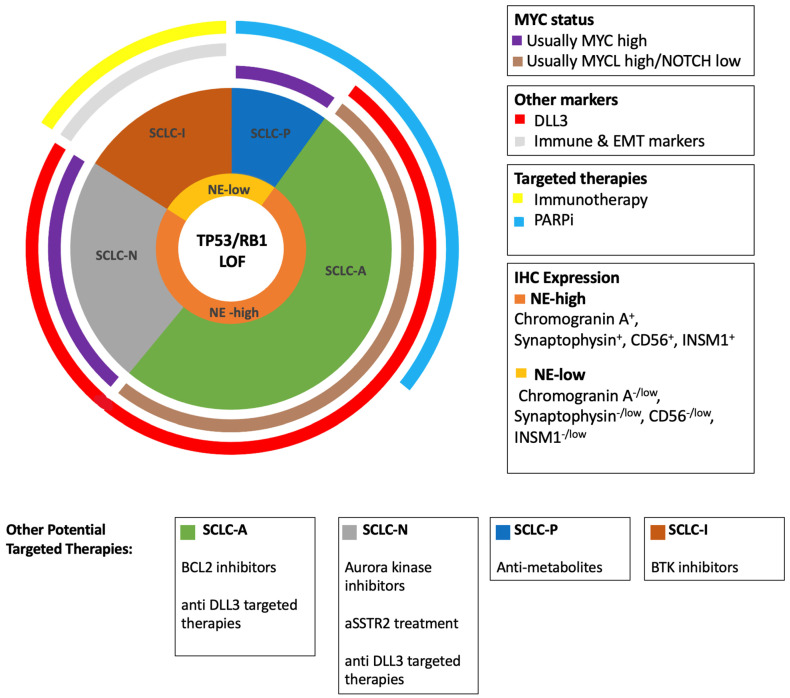
Chart of the relative abundance, MYC and Notch status, NE status, and promising targeted therapies in the four molecular subtypes of SCLC, each identified by their key transcriptional regulator. Loss of function (LOF) of RB1 and TP53 is obligatory in SCLC.

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
