# Peer review of "Emerging Biomarkers and the Changing Landscape of Small Cell Lung Cancer"

_cancers, 2022, doi:10.3390/cancers14153772_

Round 1

Reviewer 1 Report

Some Minor Corrections:

Line 22: Check “Gnomically” if the spelling is correct.

Line 24: What is “RB” kindly give abbreviation here in bracket.

Line 148: Cyclin spelling. Should be “Cyclin”.

The manuscript is well written, but I suggest authors to provide a abbreviations list and also should check spelling mistakes throughout the manuscript with some corrections which I found out. 

Author Response

Dear Reviewer, 

Thank you for your valuable comments for which we took on board as follows: 

  1. We corrected the spellings added an abbreviation for RB as suggested. We have amended any spelling errors there was throughout the paper.
  2. We added in an abbreviations list at the end of the review article as suggested, thank you. 

Reviewer 2 Report

The present manuscript reviews the emerging biomarkers for SCLC. The manuscript is well-written (mostly) and discusses several important aspects of the subject. A more structured revised version with some additional information in a language accessible to non-specialist readers would substantially improve the value and impact of this article. Any revision of the manuscript should address the comments listed below.

Comments:

1.    Authors should cite the figures in figure legends if it is not their data.

2.    Authors are required to restructure the introduction part for a better flow of reading.

3.    The authors should highlight the challenges with SCLC treatment and management compared to NSCLC where immune-based therapies are more effective.

4.    Please add the references for every scientific statement mentioned in the review article. For example, Line 158-159 should have a reference at the end.

5.    Authors should also discuss the therapeutic advantages of the predicted biomarkers for SCLC. For instance, merely highlighting the correlation is not sufficient.

6.    Please add the relevant research article instead of referring to the review article wherever it is required as it takes the credit from the actual scientific addition.

Author Response

Thank you for your valuable comments for which we took on board as follows: 

  1. References from data collected in Table one is references in line 166
  2. Thank you for your comment on improving the flow of the introduction. We altered the structure of the introduction to make it flow better. We have highlighted the sections modified (lines 49-54, 64-68)
  3. Thank you for your comment on challenges with SCLC treatment and management compared to NSCLC where immune-based therapies are more effective. We have added in a section on this and highlighted it in lines 514-528.
  4. Thank you for highlighting references for every scientific statement mentioned in the review article: we have added in references for highlighted in the article namely on line 154 (ref13), line 562 (ref 83, 84, 86-88), line 565 (ref 89), line 580 (ref 90). We have also cited original papers as suggested in comment 6.
  5. Thank you for your comment on discussing therapeutic advantages of clinical biomarkers, we have added in a comment on line 498-501 to address this.

Reviewer 3 Report

The reviewer thinks this is a well-written review article about SCLC subtypes and treatments. The reviewer raises several comments as summarized below.

1. The authors describe that "The validation of SCLC-Y (YAP-1 high) as a distinct subtype, however, failed in multiple recent studies using IHC and single cell sequencing(10, 40). Most probably the high YAP1 expression originated from inflammatory cells and NSCLC component in combined SCLCs in these early studies(40)." However, in the Figure 3, there are some YAP-1 cells and they are originated from SCLC-A (Figure 3A) but not combined SCLCs (Figure 3C). Therefore, the reviewer thinks these inconsistency should be corrected.

2. The reviewer cannot understand why the Table 1 does not have MYC, MYCL, or MYCN.

3. Regarding the SCLC transformation in lung adenocarcinomas with EGFR mutation after TKI treatment failure, the authors described that "but RB1 loss is only seen in the SCLC component". However, the reviewer thinks this is not correct in many of the cases. Both of the TP53 mutation and RB1 loss are often seen in EGFR-mutated lung adenocarcinomas and many of these tumors experience SCLC transformation after EGFR-TKI treatment failure.

4. SCLC marker status (Chromogranin A, Synaptophysin, CD56 and INSM1) should be added in the Figure 4. It would be important to know the correlation between these markers and the subtypes of SCLCs.

Author Response

Thank you for your valuable comments for which we took on board as follows: 

  1. Thank you for your comment on YAP1. We added in a new section explaining YAP1 expression in SCLC (section 5.3, lines 318-327). In figure 3C we did not add in YAP1 in the combined SCLC as current studies have identified SCLC-P as a truly distinct subtype which rarely co-expresses NEUROD1, ASCL1, or YAP1. Associations with combined SCLC and SCLC-P have also been made. We believe we have rectified the inconsistency by also adding in a description in figure 3C "SCLC-P is a distinct subgroup of SCLC rarely co-expressing other subtypes or YAP1. " (line 419) We have also referenced the papers who have identified SCLC-P to be a distinct subgroup.
  2. Thank you for your comment on Table 1. We have added in MYC family genes and other genes that should be added(line 166).
  3. Thank you for highlighting RB1 loss is seen in both SCLC and lung adenocarcinoma components of combined SCLC. We have corrected this (line 186-191) and referenced the appropriate study. 
  4. Thank you for your comment on adding in NE marker status in Figure 4. We have added this in (line 507).